# A hypothesis - generating Swedish extended national cross-sectional family study of multimorbidity severity and venous thromboembolism

Jonatan Ahrén [ORCID],[1] MirNabi Pirouzifard,[1] Björn Holmquist [ORCID],[2] Jan Sundquist,[1] Anders Halling,[1] Kristina Sundquist,[1] Bengt Zöller[1]

[1]Center for Primary Health Care Research, Department of Clinical Sciences, Lund University/ Region Skåne, Malmö, Sweden
[2]Department of Statistics, Lund University, Lund, Sweden

**Correspondence to**
Jonatan Ahrén;
Jonatan.Ahren@med.lu.se

## ABSTRACT

**Objectives** Venous thromboembolism (VTE) is a common worldwide disease. The burden of multimorbidity, that is, two or more chronic diseases, has increased. Whether multimorbidity is associated with VTE risk remains to be studied. Our aim was to determine any association between multimorbidity and VTE and any possible shared familial susceptibility.

**Design** A nationwide extended cross-sectional hypothesis - generating family study between 1997 and 2015.

**Setting** The Swedish Multigeneration Register, the National Patient Register, the Total Population Register and the Swedish cause of death register were linked.

**Participants** 2 694 442 unique individuals were analysed for VTE and multimorbidity.

**Main outcomes and measures** Multimorbidity was determined by a counting method using 45 non-communicable diseases. Multimorbidity was defined by the occurrence of ≥2 diseases. A multimorbidity score was constructed defined by 0, 1, 2, 3, 4 or 5 or more diseases.

**Results** Sixteen percent (n=440 742) of the study population was multimorbid. Of the multimorbid patients, 58% were females. There was an association between multimorbidity and VTE. The adjusted odds ratio (OR) for VTE in individuals with multimorbidity (2 ≥ diagnoses) was 3.16 (95% CI: 3.06 to 3.27) compared with individuals without multimorbidity. There was an association between number of diseases and VTE. The adjusted OR was 1.94 (95% CI: 1.86 to 2.02) for one disease, 2.93 (95% CI: 2.80 to 3.08) for two diseases, 4.07 (95% CI: 3.85 to 4.31) for three diseases, 5.46 (95% CI: 5.10 to 5.85) for four diseases and 9.08 (95% CI: 8.56 to 9.64) for 5 ≥ diseases. The association between multimorbidity and VTE was stronger in males OR 3.45 (3.29 to 3.62) than in females OR 2.91 (2.77 to 3.04). There were significant but mostly weak familial associations between multimorbidity in relatives and VTE.

**Conclusions** Increasing multimorbidity exhibits a strong and increasing association with VTE. Familial associations suggest a weak shared familial susceptibility. The association between multimorbidity and VTE suggests that future cohort studies where multimorbidity is used to predict VTE might be worthwhile.

## STRENGTHS AND LIMITATIONS OF THIS STUDY

⇒ High coverage and high validity of the registers and the large number of unique individuals.
⇒ Only Swedish-born individuals born by Swedish parents were included. It therefore remains for a future study to show if the results may be generalised to other populations.
⇒ The data did not include biological and environmental risk factors, to make up for the absence of such data we adjusted for educational level, which is related to lifestyle factors.
⇒ Due to the cross-sectional design, a temporal and possible causal association remains to be determined.

## INTRODUCTION

Venous thromboembolism (VTE) is the third most common cardiovascular disease with an incidence of around 1–3 per 1000 individuals.[1–4] The incidence rate increases with age. VTE is rare among children with an incidence of approximately 1 per 100 000 and common among elderly people >70 years and above with an incidence of around 1% per year.[5] The lifetime risk of VTE could be up to 10% in certain countries, for instance in Sweden.[3 4] VTE usually manifests as deep venous thrombosis (DVT) of the lower extremities or as pulmonary embolism (PE).[6] The consequences of VTE range from post-thrombotic syndrome, which affects approximately 20% of patients with DVT, to acute death in 1%–2% of PE patients.[7 8] The 90 day mortality after VTE is around 5%.[9]

Risk factors for VTE are either acquired or genetic but they often interact.[8] Among the genetic risk factors, the most prominent are the classical thrombophilia, that is, factor V Leiden (rs6025), the prothrombin G20210A mutation (rs1799963), deficiencies of antithrombin, protein C and protein S.[10 11] Among the acquired risk factors, age

is the most significant with a steeply increased risk with advanced age.[5][6] Other acquired risk factors are for instance immobilisation, trauma and surgery, malignancy, pregnancy and puerperium, oral contraception and auto-immune disorders.[1][2][6][12][13]

Risk factors for VTE may work alone or in concert.[6][12] Multimorbidity, that is, the presence of two or more diseases, increases with age.[14] As patients get older and the field of medicine advances the patients more often present with multiple diseases—multimorbidity.[14] However, multimorbidity also occurs among younger persons.[15] Genetic inheritance has been implicated in multimorbidity.[16] Dong et al have studied 439 common diseases among 385355 patients in UK biobank. They identified 11285 multimorbidities and using genome-wide association study data, they showed that 46% of multimorbidities were genetically interpretable (https://multimorbidity.comp-sysbio.org).[16] Due to this, there is a need to investigate multimorbidity as a risk factor for additional illnesses, such as VTE which also have a genetic background. With the future demographic change with an increasingly elderly population with multimorbidity, there may be a need to adopt a multidisease framework for clinical diagnostics and treatment. Despite the need to characterise multimorbidity, there is no established definition of which morbidities should be included to define multimorbidity.[17] Multimorbidity has often been defined as two or more chronic non-communicable diseases (NCD).[17][18] Barnett et al used 40 diseases in a Scottish cross-sectional study and defined multimorbidity as the presence of two or more diseases where the prevalence of multimorbidity was 23.2% and was present in most people 65 years of age and older.[14] Although the frequency of multimorbidity increases with age, the burden of multimorbidity may be more common in those aged 65 years or younger, and this is especially common in socially deprived neighbourhoods.[19] Several NCDs, but not all, included in the multimorbidity study by Barnett et al[14] have been associated with VTE.[20–39] For some associations between NCDs and VTE, the associations are bidirectional,[36–38] for example, atherosclerotic disease and atrial fibrillation, but for most NCDs, the association with VTE is unidirectional.[20–35] However, to the best of our knowledge, the association between multimorbidity and its severity (ie, number of NCDs) and VTE has not been studied.

We aimed to determine whether there is an association between multimorbidity and its severity (ie, number of NCDs) and VTE. We used large Swedish national registers to study the association between multimorbidity and VTE. A modified list of 45 NCDs diagnoses used by Barnett et al was used to define multimorbidity as the presence of two or more of 45 NCDs.[14] To determine a possible association between multimorbidity and VTE, we also examined the association between the number of diseases and VTE. We also investigated whether a familial association of multimorbidity and VTE exists. Familial associations of a disease may be due to shared environmental exposure(s), shared genetic susceptibility or a combination of the two,[40] but familial association of a disease is a necessary, though insufficient, condition to infer the importance of genetic susceptibility.[40]

## MATERIAL AND METHODS
### Registers
The following registers were used in the study: The Swedish Multigeneration Register contains data of familial relationships; the National Patient Register (NPR) contains hospital discharge diagnoses between 1964 and 2015 with nationwide coverage from 1987 and from 2001 to 2015 outpatient hospital diagnoses; the Total Population Register with data on death date, education, marital status, migration and the Swedish cause of death register.[41–46] We used the unique Swedish personal identity number (PIN) to link the databases. The PIN was, however, replaced with a serial number to ensure confidentiality by Statistics Sweden. The validity of Swedish registers has been investigated previously and they have a high degree of coverage and validity.[41–46] Statistics Sweden and the National Board of Health and Welfare provided the register data to us.

### Study design
A nationwide extended cross-sectional study between 1997 and 2015. This period was chosen to increase the coverage of the registers and to use only the latest versions of the international classification of disease (ICD) 10 codes. To minimise the risk of an incorrect PIN, only Swedish-born individuals were included.[42] To investigate any possible familial aggregation, we included pairs of full siblings born between 1948 and 2005 by Swedish parents who could further be linked to their Swedish-born half-siblings, and cousins to minimise this source of error. Parents in the study were born between 1932 and 1985 in Sweden. The Swedish National Multigeneration Register was used to link relatives. We excluded the families with members who died or emigrated before 1997 or emigrated before the age of 17. Both biological parents were obligatorily known, that is, pedigrees with two parents and two children. Full siblings were thereafter linked with related families and with the full siblings' half-siblings and cousins. Full siblings will hereafter be referred to as 'siblings'. One data set with all individuals entered only once was created. Four data sets for familial relations were created: twins, siblings, half-siblings and cousins. In the four familial data sets, all relative pairs were double-entered, that is, all sibling pairs, all twins' pairs, all half-sibling pairs and all cousin pairs, as described previously.[47] The double entry approach is a common procedure in genetics.[47][48] Zygosity was assessed only from sex, that is, twins with different sex were considered to be DZ twins. We had no access to zygosity data. We allowed the same person to be included in more than one family relationship.

### Definition of venous thromboembolism
Cases of VTE (ie, affection status) were classified according to the WHO ICD, Tenth Revision (ICD-10).

VTE was defined using the hospital discharge register, main and supplementary diagnoses (between 1997 and 2015) by the following ICD codes: ICD-10: I80 (not I80.0) and I26.[11] It was also defined in the Outpatient Care Register (between 2001 and 2015) by the same ICD-10 codes as above and the cause of death register. Thus, VTE included PE and deep vein thrombosis of the lower extremities. The diagnosis of VTE in the NPR has been shown to have an accuracy of 95%, whereas the overall validity of the NPR is 87%.[43 48] A quality control of the NPR register was performed for 118 patients with VTE in the Malmö diet and cancer study cohort. In 106 (90%) of 118 cases the diagnosis of VTE was correct.[4]

### Multimorbidity score

Since there is no clear definition of which NCDs should be included, the NCDs used in Barnett *et al* were used and modified to adapt to the Swedish ICD-10 codes (online supplemental table S1). Further modifications were made to the list and infectious diseases were excluded. Five other clinically important and common NCDs contributing to disease burden were added instead: pancreatitis, obesity, gout, arthrosis and osteoporosis. Furthermore, psoriasis and dermatitis or eczema were counted separately. These modifications were made to increase the number of NCDs to get a more comprehensive list. Thus, a total of 45 diagnoses were included and an individual with two or more NCDs was considered to have multimorbidity. Each diagnosis in an individual during the study period was awarded one point per disease and was cumulatively counted. An individual could theoretically have 45 points. In the study, no individual obtained more than 20 points.

### Statistical analysis

Logistic regression was used to investigate the association between VTE and multimorbidity. Multimorbidity was defined as two or more diseases. Multimorbidity was also investigated as a cumulative score in order to determine any possible increasing association. Results are presented as Odds Ratios (ORs) with 95% confidence intervals (CIs). Adjustments were made for year of birth, sex, region at birth and level of education. Stratified analysis was performed according to sex and birth year quartiles. The familial associations were determined using the double-entry approach.[47] The ambiguity in unselected samples as to which twin's, sibling's, half-sibling's or cousin's trait should be used as the dependent, and which as the independent variable, could be resolved by using double-entry.[47] Each twin, sibling, half-sibling or cousin is entered twice in the data, and each member of a twin, sibling, half-sibling or cousin pair provides once the dependent and once the explanatory variable.[47] While the consistency of the regression estimates for heritability and environmental influences is not affected by double-entry, the SEs of the coefficients are biased and need to be adjusted.[47] In the present study in the logistic regression models, we used the 'variance covariance (vce) cluster'

method using STATA, which calculated robust SEs using families as clusters.[49] The vce (cluster clustvar) specifies that the SEs allow for intragroup correlation, relaxing the usual requirement that the observations be independent.

Statistical significance was set to $p < 0,05$, all tests were two-tailed. Data were analysed using STATA V.16.1 (StataCorp LLC, College Station, TX, USA)

### Patient and public involvement
None.

## RESULTS
### Multimorbidity

A total of 2 694 442 unique individuals were included in the study, 49% of these were females. The mean age at the end of the study was 32 years. During the study period between 1997 and 2015, a total of 440 742 (16.36%) individuals were diagnosed with multimorbidity (table 1). Of these, 253 931 (57.61%) were females and the proportion of females increased with increasing multimorbidity score with 63.27% of the individuals with a score of ≥5 being female (online supplemental table 2). Multimorbidity was associated with level of education where 25.47% of the multimorbid individuals had an education of ≥12 years compared with 31.32% in the non-multimorbid group (table 1). Multimorbidity also increased with older age at the end of the study, multimorbid individuals had a mean age of 35 years versus 31 in non-multimorbid groups. Baseline characteristics can be found in table 1 and in online supplemental table 2.

### Venous thromboembolism

There was a total of 16 099 cases with diagnosed VTE during the study period between 1997 and 2015. In individuals with multimorbidity, 1.58% had VTE compared with 0.40% in individuals without multimorbidity. Of the female individuals, 0.66% had VTE and among males, the figure was 0.54%. VTE increased with age with 1.3% in the oldest group and 0.08% in the youngest group. (tables 1–3).

### VTE and multimorbidity

In the multimorbidity group, a total of 6983 of the 440 742 (1.58%) individuals were diagnosed with VTE (table 1). In the group with a multimorbidity score of ≥5, 1603 of the 36 313 (4.41%) had VTE during the study period (online supplemental table 2). The OR for VTE in the adjusted model for an individual with multimorbidity was 3.16 (95% CI: 3.06 to 3.27) (table 2). For males with multimorbidity, the OR for VTE was higher than for females (table 2). Birth year was stratified into quartiles. For individuals born between 1947 and 1972, the OR for VTE with multimorbidity was 3.14 (95% CI: 3.01 to 3.29) in the adjusted model and for individuals born between 1993 and 2005, the OR for VTE with multimorbidity was 3.07 (95% CI: 2.56 to 3.7) (table 3). Thus, there was no

**Table 1** Descriptive findings for all unique study participants with stratification according to multimorbidity scores for number of unique individuals, sex, education, age at the end of study, birth date and VTE (venous thromboembolism)

| | All | Multimorbidity scores* | |
| | | ≤1 | ≥2 |
|---|---|---|---|
| Unique individuals, % (n) | 100% (n=2 694 442) | 83.64% (n=2 253 700) | 16.36% (n=440 742) |
| Sex, female, % (n) | 48.73% (n=1 312 989) | 80.66% (n=1 059 058) | 19.34% (n=253 931) |
| Education (≥12 years), % (n) | 30.36% (n=818 146) | 86.28% (n=705 901) | 13.72% (n=112 245) |
| Age at end of study (years), median (IQR) range (min–max) | 32 (22–43) (0–68) | 31 (22–42) (0–67) | 35 (24–46) (0–68) |
| Year of birth, median (IQR) range (min–max) | 1983 (1972–1993) (1947–2005) | 1984 (1973–1993) (1948–2005) | 1980 (1969–1990) (1947–2005) |
| VTE, % (n) | 0.6% (n=16 099) | 56.62% (n=9116) | 43.38% (n=6983) |

*%=For multimorbidity scores, percentage is calculated from the total number of people given in the first column (all columns).
IQR, Interquartile Range; n, number of individuals.

strong age dependence regarding the association between multimorbidity and VTE.

### Association between increasing multimorbidity score and VTE
The association of increasing multimorbidity score on VTE was investigated (table 4). It showed an increased OR for increasing score of multimorbidity. OR for a multimorbidity score of two in the adjusted model was 2.93 (95% CI: 2.80 to 3.08), 4.07 (95% CI: 3.85 to 4.31) for a multimorbidity score of three, 5.46 (95% CI: 5.10 to 5.85) for a multimorbidity score of four and for a multimorbidity score of five or more, the OR was 9.08 (95% CI: 8.56 to 9.64) (table 4).

### Age and sex stratification
Multimorbidity score results for sex and age stratification are presented in online supplemental tables S3–S8. A strong association between multimorbidity and VTE could be observed in both sexes and all age groups.

### Familial associations between multimorbidity and VTE
There was mostly a weak association between twin, sibling, half-sibling and cousin history of multimorbidity and VTE (table 5). The ORs in adjusted models for VTE in patients with multimorbidity were 1.89 in twins (95% CI: 1.17 to 3.05), 1.18 in siblings (95% CI: 1.12 to 1.24), 1.1 in half-siblings (95% CI: 1.03 to 1.18) and 1.07 in cousins (95% CI: 1.04 to 1.1) of those diagnosed with index score more than one (table 5). Among twins, the adjusted OR for VTE in patients with multimorbidity was=0.73 (95% CI: 0.04 to 1.35) for twin score=1, OR=1.57 (95% CI: 0.81 to 3.05) for twin score=2, OR=1.26 (95% CI: 0.48 to 3.31) for twin score=3, OR=1.48 (95% CI: 0.44 to 4.98) for twin score=4 and OR=3.98 (95% CI: 1.17 to 9.39) for twin score ≥5.

### DISCUSSION
The strong association between multimorbidity and VTE is, to the best of our knowledge, a novel finding. However, the study design does not allow any causal inferences to

**Table 2** Odds Ratio (OR) for VTE (venous thromboembolism) and multimorbidity (score ≥2) for all and stratified according to sex

| | OR (95% CI) | | | |
| | No VTE | VTE | Model 1 | Model 2 |
|---|---|---|---|---|
| All | 433 759 | 6983 | 3.96 (3.84 to 4.09) | 3.16 (3.06 to 3.27) |
| Males | 183 732 | 3079 | 4.52 (4.32 to 4.74) | 3.45 (3.29 to 3.62) |
| Females | 250 027 | 3904 | 3.5 (3.35 to 3.65) | 2.91 (2.79 to 3.04) |

Total observations n=2 694 442.
Model 1 is a crude model (univariable). Model 2 is an adjusted model (multivariable), with adjustments for year of birth, county and educational attainment.
ORs with 95% CI for multimorbidity. Reference with one or no diseases (score=0 or 1).
CI, Confidence Interval.

**Table 3** Odds Ratio (OR) for VTE (venous thromboembolism) according to multimorbidity (score ≥2) and age stratified according to Interquartile Range (IQR1–4)

| Age groups | OR (95% CI) | | | |
| --- | --- | --- | --- | --- |
| | No VTE | VTE | Model 1 | Model 2 |
| IQR1 | 135 989 | 4032 | 3.35 (3.21 to 3.50) | 3.14 (3.01 to 3.29) |
| IQR2 | 106 001 | 1857 | 3.43 (3.24 to 3.64) | 3.14 (2.96 to 3.34) |
| IQR3 | 104 897 | 919 | 3.48 (3.21 to 3.78) | 3.09 (2.84 to 3.37) |
| IQR4 | 86 872 | 175 | 3.52 (2.93 to 4.22) | 3.07 (2.56 to 3.7) |

ORs with 95 % CI for multimorbidity scores. Reference with no diseases (score=0). Total observations n=2 694 442.
Model 1 is a crude model (univariable). Model 2 is an adjusted model (multivariable), with adjustments for sex, year of birth, county and educational attainment.
IQR1: 1947–1972, IQR2: 1972–1983, IQR3: 1983–1993 and IQR4: 1993–2005.
CI, Confidence Interval.

be drawn regarding multimorbidity and VTE. Still, the strong and increasing association with increasing multimorbidity score suggests that future cohort studies might be considered in order to determine whether a possible causal association between multimorbidity and VTE exists.[50] Our present study shows an association between multimorbidity and VTE with an adjusted OR of 3.16. The VTE risk increases with a higher multimorbidity score and is nine times higher in individuals with a multimorbidity score of five or more compared with individuals with no morbidity. Thus, multimorbidity is strongly associated with VTE. Genetic inheritance has been implicated in multimorbidity.[16] However, our data argue against a major genetic association between multimorbidity and VTE due to the mostly weak familial associations. However, there was a significant familial association, which correlated with the degree of relatedness suggesting a possible shared genetic susceptibility to the association between multimorbidity and VTE, though shared environmental exposures also may be involved. A high multimorbidity score in a patient confers similar ORs for VTE as strong genetic or acquired risk factors.[1 2 6 10–12] Moreover, multimorbidity has been associated with inflammatory

biomarkers and an association between inflammation and VTE has long been recognised.[51–54] Based on these findings, future dissection of multimorbidity and its relation to VTE in cohort studies could be of value.

The impact of multimorbidity on VTE is stronger in males compared with females and with a higher relationship among males. On the other hand, females have increased multimorbidity and live longer which may give a double burden for the female population in the context risk of VTE, if multimorbidity will turn out to be an important risk factor for VTE.

The risk of multimorbidity is increasing with age and more individuals worldwide are multimorbid.[16] Our study population is relatively young and confirms previous studies that multimorbidity may also affect young persons.[15] However, there was a strong association with multimorbidity in all age groups suggesting that multimorbidity might also be of importance in older age groups not studied here; this needs to be confirmed in further studies. If multimorbidity in future studies turn out to be a risk factor for VTE, it is possible that multimorbidity is an important contributor to the high incidence of VTE in society due to its high prevalence. Common risk factors,

**Table 4** Odds ratio (OR) for VTE (venous thromboembolism) according to multimorbidity score (0 to ≥5)

| Multimorbidity score | All (n=2 694 442) | | | |
| --- | --- | --- | --- | --- |
| | OR (95% CI) | | | |
| | No VTE | VTE | Model 1 | Model 2 |
| Score 0 | 1 611 205 | 5008 | 1 (reference) | 1 (reference) |
| Score 1 | 633 379 | 4108 | 2.09 (2.00 to 2.18) | 1.94 (1.86 to 2.02) |
| Score 2 | 251 558 | 2652 | 3.39 (3.24 to 3.56) | 2.93 (2.80 to 3.08) |
| Score 3 | 102 969 | 1665 | 5.20 (4.92 to 5.50) | 4.07 (3.85 to 4.31) |
| Score 4 | 44 522 | 1063 | 7.68 (7.18 to 8.21) | 5.46 (5.10 to 5.85) |
| Score ≥5 | 34 710 | 1603 | 14.86 (14.03 to 15.73) | 9.08 (8.56 to 9.64) |

ORs with 95 % CI for multimorbidity scores. Reference with no diseases (score=0).
Model 1 is a crude model (univariable). Model 2 is an adjusted model (multivariable), with adjustments for sex, year of birth, county and educational attainment.
CI, Confidence Interval.

**Table 5** Familial associations for multimorbidity and venous thromboembolism (VTE)

| | Twins (n=24 020) | | Siblings (n=1 546 108) | | Half-siblings (n=984 976) | | Cousins (n=6 623 156) | |
|---|---|---|---|---|---|---|---|---|
| | OR (95% CI) | | OR (95% CI) | | OR (95% CI) | | OR (95% CI) | |
| VTE risk in individuals with relatives with score between 1 and ≥5. | | | | | | | | |
| Relative score | Model 1 | Model 2 | Model 1 | Model 2 | Model 1 | Model 2 | Model 1 | Model 2 |
| Score 0 | Reference | Reference | Reference | Reference | Reference | Reference | Reference | Reference |
| Score 1 | 0.77 (0.42 to 1.41) | 0.73 (0.4 to 1.35) | 1.19 (1.13 to 1.25) | 1.11 (1.06 to 1.17) | 1.12 (1.05 to 1.19) | 1.08 (1.01 to 1.15) | 1.09 (1.06 to 1.12) | 1.04 (1.02 to 1.07) |
| Score 2 | 1.81 (0.95 to 3.46) | 1.57 (0.81 to 3.05) | 1.37 (1.28 to 1.46) | 1.19 (1.11 to 1.27) | 1.14 (1.05 to 1.25) | 1.07 (0.98 to 1.17) | 1.16 (1.11 to 1.2) | 1.07 (1.03 to 1.11) |
| Score 3 | 1.86 (0.74 to 4.68) | 1.26 (0.48 to 3.31) | 1.5 (1.37 to 1.67) | 1.18 (1.07 to 1.3) | 1.35 (1.2 to 1.51) | 1.19 (1.07 to 1.34) | 1.21 (1.14 to 1.27) | 1.06 (1 to 1.11) |
| Score 4 | 2.66 (0.82 to 8.59) | 1.48 (0.44 to 4.98) | 1.89 (1.67 to 2.15) | 1.34 (1.18 to 1.53) | 1.4 (1.19 to 1.64) | 1.17 (1.0 to 1.37) | 1.38 (1.28 to 1.48) | 1.14 (1.06 to 1.22) |
| Score ≥5 | 9.31 (4.17 to 20.79) | 3.98 (1.69 to 9.39) | 2.16 (1.88 to 2.48) | 1.31 (1.14 to 1.51) | 1.6 (1.36 to 1.89) | 1.2 (1.02 to 1.41) | 1.52 (1.41 to 1.65) | 1.16 (1.07 to 1.26) |
| VTE risk in individuals with relatives with multimorbidity (score ≥2) | | | | | | | | |
| Score <2 | Reference | Reference | Reference | Reference | Reference | Reference | Reference | Reference |
| Score ≥2 | 2.55 (1.62 to 4.02) | 1.89 (1.17 to 3.05) | 1.43 (1.36 to 1.51) | 1.18 (1.12 to 1.24) | 1.23 (1.15 to 1.31) | 1.1 (1.03 to 1.18) | 1.19 (1.16 to 1.23) | 1.07 (1.04 to 1.1) |

OR with 95% CI for VTE according to multimorbidity score (0 to ≥5) or multimorbidity (≥2 score) in relatives. References are individuals with relatives with no diseases (score=0) for the graded score (score between 1 and ≥5). For multimorbidity (score ≥2), references are individuals with relatives with no multimorbidity (score <2), that is, no or one disease. Zygosity of twins are not known. The ORs were derived from double entry and the variance covariance (vce) cluster method was used. Model 1 is a crude model. Model 2 is adjusted for sex, year of birth, county and educational attainment.

CI, Confidence Interval; OR, Odds Ratio.

both genetic and acquired, are often seen as individual risk factors in specific settings, like surgery, pregnancy, puerperium, oral contraceptives, surgery and trauma as well as for specific diseases, such as cancer. The present study suggests that we might count the total burden of disease into a multimorbidity score to get a better risk estimate of VTE. However, future prospective studies need to determine which combinations of diseases are most strongly associated with VTE.

Since there is no clear definition of which diagnoses should be used in the field of multimorbidity, the ones that are included are sometimes random.[14–19] In this study, we used a modified score validated by Barnett *et al*.[14] The NCDs that were used in this study are all common. They all have an impact on healthcare, and most are considerably stable during an individual's life and can coexist and create a significant health burden.[14] It will be important to investigate whether a certain combination of diagnoses and multimorbidity has an especially strong association with VTE and further studies are needed to investigate this matter. However, the association between multimorbidity and VTE suggests that both specific diseases are important and also the total burden of certain diseases.

## Strengths and limitations

The limited number of confounders is a potential limitation. An important limitation of the study is that we did not determine any temporal association between multimorbidity and VTE. We therefore cannot draw any causal inferences or exclude reverse causality. However, only atrial fibrillation and arterial atherosclerotic diseases have been described to exhibit a bidirectional association with VTE.[36–38] Moreover, it is possible that certain combinations of diseases are more strongly associated with VTE than others. It will be of importance to further dissect the occurrence of multimorbidity and its relation to future risk for VTE in cohort studies. The study period was chosen to minimise the risk of missed data and to increase the coverage of individuals. Furthermore, having more comprehensive ICD-10 codes decreases the risk of inaccurate coding from the registers. A strength of the study is the high coverage and high validity of the registers and the large number of unique individuals.[41–46] A limitation is that only Swedish-born individuals born by Swedish parents were included. It therefore remains for a future study to show if the results may be generalised to other populations. Another limitation is that the data did not include biological and environmental risk factors such as smoking, alcohol consumption, height, weight, blood pressure and exercise habits. To make up for the absence of such data, we adjusted for educational level, which is related to lifestyle factors.[55] Also, the included patients were relatively young with an age-range of 0–68 and a mean of 32 at the end of the study and generalisation of the elderly is uncertain, even though within this age span between 0 and 68 years, there was no age effect on the association. Hence, there is a need for further studies in the future.

## CONCLUSION

We have shown that an association exists between multimorbidity and multimorbidity severity (ie, number of NCDs) and VTE. Weak shared familial factors (environmental or genetic) may also be involved. The relation between multimorbidity and VTE could be further studied in cohort studies and Mendelian randomization studies in order to determine a temporal and possibly causal association.

**Acknowledgements** We thank Patrick Reilly, science editor at Centre for Primary Health Care Research, Department of Clinical Science, Lund University and Region Skåne, Malmö, Sweden, for language review.

**Contributors** JA, MP, BH, JS, AH, KS and BZ contributed to the conception and design of the study, analysis and interpretation of data; JS and KS contributed to the acquisition of data; JA drafted the manuscript; JA, MP, BH, JS, AH, KS and BZ revised it critically and approved the final version. JA, MP, BH, JS, AH, KS and BZ had full access to all the data (including statistical reports and tables) and take responsibility for the integrity of the data and the accuracy of their analysis.

**Funding** ALF-funding from Region Skåne (N/A) and The Swedish Research Council (2020-01824).

**Competing interests** None declared.

**Patient and public involvement** Patients and/or the public were not involved in the design, or conduct, or reporting or dissemination plans of this research.

**Patient consent for publication** Not required.

**Ethics approval** This study was approved by the Regional Ethical Review Board in Lund (2012/795 and later amendments).

**Provenance and peer review** Not commissioned; externally peer reviewed.

**Data availability statement** Data may be obtained from a third party and are not publicly available. We are not allowed to share our data as the data are drawn from registries owned by a third party, that is, the Swedish authorities. The reason behind this is that public availability could potentially compromise patient confidentiality or privacy. Permission to use the data is issued to researchers (if certain conditions are fulfilled) by the National Board of Health and Welfare and by Statistics Sweden. Request to use the data can be applied for from the Swedish National Board of Health and Welfare (https://www.socialstyrelsen.se/en/statistics-and-data/statistics/statistical-database/) and Statistics Sweden (https://www.scb.se/en/About-us/contact-us/). An application for use of the same minimal data set, which we have used in this study, should include a motivation, a referral to the present study and an addition of a list of variables, before submission of a data request to the Swedish National Board of Health and Welfare and Statistics Sweden using the contact addresses above. Then, the authorities will coordinate the handling of the request and do a special review.

**ORCID iDs**
Jonatan Ahrén http://orcid.org/0000-0003-1307-5055
Björn Holmquist http://orcid.org/0000-0003-4316-2483

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
