## [Reviewer comments · BMJ Open]

ARTICLE DETAILS

TITLE (PROVISIONAL)	A hypothesis - generating Swedish extended national cross-sectional family study of multimorbidity severity and venous thromboembolism
AUTHORS	Ahrén, Jonatan; Pirouzifard, Mirnabi; Holmquist, Björn; Sundquist, Jan; Halling, Anders; Sundquist, Kristina; Zöller, Bengt

VERSION 1 – REVIEW

REVIEWER	Grainge, Matthew University of Nottingham, Division of Epidemiology and Public Health
REVIEW RETURNED	03-Mar-2023

GENERAL COMMENTS	In this paper, Ahren and colleagues used linked nationwide healthcare data from Sweden to explore the association between multimorbidity and risk of venous thromboembolism (VTE). Strengths of the study include the large sample size, quality of data recording and validation of the electronic coding of VTE. One of key messages, both in the summary boxes and discussion, was to map which diseases contribute most to the increased risk of VTE. However, I am not sure how this would add to the body of knowledge on this topic because there is already a large volume of literature for individual medical conditions (e.g. cancer and IBD) on the risk of VTE. My own view is that this study provides added value as it explores whether for purposes of VTE risk stratification, a simple count of number of comorbid conditions would be as predictive as considering the conditions separately. The disadvantage of loss of information by not weighting medical conditions by strength of association could be more than offset by the simplicity of a comorbidity count. However, this is something which is not addressed in either the analysis or discussion of findings. My belief is that this work would be strengthened by citing (and critically evaluating where necessary) previous systematic reviews which have evaluated the relationship between individual NCDs and risk of VTE. This would allow the reader to assess the potential added value of considering use of a count of comorbidities for the purpose of risk stratification. My other concern about the paper surrounds the dates of data recording. The authors should define clearly over what period of time comorbidities were assessed (for instance some comorbidity scores count only diagnoses in a fixed period time, say the year leading up to an event). If medical diagnoses or VTE could occur at any time in the interval 1987 to 2015 then some comorbidities could be coded after the outcome of VTE. This raises the concern over reverse causality as a blood clot could directly increase the risk of having some of these NCDs developing. This would limit the
--

	value of these results, if the purpose is to use counts of NCD as part of VTE risk stratification in clinical practice. Related to the above comment, odds ratios are used as the analysis measure which is more usual for case-control than cohort studies. If the above concerns over timing of events can be addressed then a time to event analysis which takes into account the time under observation (and therefore likelihood of developing VTE) could be employed. Other comments: The limited number of confounders needs to be highlighted more clearly as a potential limitation. As acknowledged, education level was considered as a proxy for lifestyle factors. However, if lifestyle factors rather than the NCDs themselves were the reason for the association, this would affect the interpretation of findings (and the potential for comorbidity count to be used within risk assessment tools). Table 1 may be easier to follow if presented as the percent of people in each comorbidity category by risk factor (i.e. for males/females, high education/low education) rather than the other way round. Please use the terms 'univariable' and 'multivariable' rather than 'univariate' and 'multivariate' throughout the paper.
--	---

REVIEWER	van Mens, Thijs E. University of Amsterdam
REVIEW RETURNED	22-Mar-2023

GENERAL COMMENTS	This is a well performed study showing a robust association between multimorbidity and VTE, using relatively unique population data. The methodology is overall valid. However, this reviewers main concern is the added value of the results to the field. The low granularity and accuracy of these data hamper the interpretability. The findings of multimorbidity, cumulative during a long period of time, being associated with VTE (at any time point during the same period, without a temporal relation), neither adds new insight in epidemiology of VTE, as it is well known that a variety of diseases are associated with VTE. Hence multimorbidity will also be associated. Nor does it have clear clinical implications. This knowledge does not translate into clinical decision in VTE care. I do not share the conclusion that “ The present study suggests that we might count the total burden of disease into a multimorbidity score to get a better risk estimate of VTE. “. In my opinion, that would be a step backwards, from predicting using well defined risk factors of which we often also (in part) understand the underlying pathophysiology, to predicting based on a reduced summary of the patient simply because these are (the low-granularity) data that are available. The research question of familial association is not supported by a clear rationale, except for an explanation that familial association is a proxy for genetic susceptibility. Detailed comments: In the multivariate models, year of birth was included. Age at diagnoses of VTE seems a more accurate variable.
---

	It appears that multimorbidity is determined over the entire study period. This further limits the interpretability, as one would be interested in multimorbidity at time of VTE diagnosis. The lack of temporal association with VTE is another major drawback, as is the lack absence of data on known life style risk factors. Both limitations are clearly considered and described by the authors.
--	--

VERSION 1 – AUTHOR RESPONSE

Reviewer: 1

Dr. Matthew Grainge, University of Nottingham

Comments to the Author:

In this paper, Ahren and colleagues used linked nationwide healthcare data from Sweden to explore the association between multimorbidity and risk of venous thromboembolism (VTE). Strengths of the study include the large sample size, quality of data recording and validation of the electronic coding of VTE. One of key messages, both in the summary boxes and discussion, was to map which diseases contribute most to the increased risk of VTE. However, I am not sure how this would add to the body of knowledge on this topic because there is already a large volume of literature for individual medical conditions (e.g. cancer and IBD) on the risk of VTE. My own view is that this study provides added value as it explores whether for purposes of VTE risk stratification, a simple count of number of comorbid conditions would be as predictive as considering the conditions separately. The disadvantage of loss of information by not weighting medical conditions by strength of association could be more than offset by the simplicity of a comorbidity count. However, this is something which is not addressed in either the analysis or discussion of findings. My belief is that this work would be strengthened by citing (and critically evaluating where necessary) previous systematic reviews which have evaluated the relationship between individual NCDs and risk of VTE. This would allow the reader to assess the potential added value of considering use of a count of comorbidities for the purpose of risk stratification.

Author response: Thank you for the appreciation of the study and for mentioning that this study provides added value. Please note that the summary boxes have been deleted as suggested by the Editor. The title has also been changed (as suggested by the Editor). We agree with the reviewer that the added value of the present study is the exploration of whether multimorbidity and a simple counting method for NCDs could be associated with VTE. We also appreciate the suggestion to add more references and have now added several systematic reviews and large observational studies. We also clearly state that, to the best of our knowledge, nobody has examined the association between multimorbidity or multimorbidity severity (i.e. number of NCDs) and venous thromboembolism (VTE). We believe that our findings could be of interest for other researchers in the field because of the hypothesis-generating nature of our study.

My other concern about the paper surrounds the dates of data recording. The authors should define clearly over what period of time comorbidities were assessed (for instance some comorbidity scores count only diagnoses in a fixed period time, say the year leading up to an event). If medical diagnoses or VTE could occur at any time in the interval 1987 to 2015 then some comorbidities could be coded after the outcome of VTE. This raises the concern over reverse causality as a blood clot could directly increase the risk of having some of these NCDs developing. This would limit the value of these results, if the purpose is to use counts of NCD as part of VTE risk stratification in clinical practice.

Author response: The reviewer is correct that there is a certain risk for reversed causation, which now is recognized. However, this would apply only for atrial fibrillation and arterial cardiovascular disorders, such as coronary heart disease, stroke and other atherosclerotic diseases, where a bidirectional association has been described. Thus, even if there is a certain risk for reverse causation, it should be unlikely to explain all of our findings. However, we now stress that this is a cross-sectional hypothesis - generating study and that cohort studies are necessary to exclude the possibility of reverse causation.

Related to the above comment, odds ratios are used as the analysis measure which is more usual for case-control than cohort studies. If the above concerns over timing of events can be addressed then a time to event analysis which takes into account the time under observation (and therefore likelihood of developing VTE) could be employed.

Author response: The reviewer is right. A cohort approach is necessary for establishing a temporal association. However, we performed the present study as a cross-sectional hypothesis - generating study and odds ratios are used when the approach is cross-sectional. To the best of our knowledge, no study has examined the association between multimorbidity or multimorbidity severity (i.e. number of NCDs) and venous thromboembolism (VTE). The present study is hypothesis - generating and may motivate future case-control and cohort studies.

Other comments:

The limited number of confounders needs to be highlighted more clearly as a potential limitation. As acknowledged, education level was considered as a proxy for lifestyle factors. However, if lifestyle factors rather than the NCDs themselves were the reason for the association, this would affect the interpretation of findings (and the potential for comorbidity count to be used within risk assessment tools).

Author response: We now mention the limited number of confounders as a potential limitation. We present both a crude model 1 and an adjusted model 2. The influence of education was limited. This, as indicated by the reviewer, suggests that the NCDs themselves rather than education and associated lifestyle factors should explain our findings.

Table 1 may be easier to follow if presented as the percent of people in each comorbidity category by risk factor (i.e. for males/females, high education/low education) rather than the other way round.

Author response: We have now revised table 1 in accordance with this comment.

Please use the terms 'univariable' and 'multivariable' rather than 'univariate' and 'multivariate' throughout the paper.

Author response: 'univariate' and 'multivariate' are now changed to 'univariable' and 'multivariable' throughout the paper.

Reviewer: 2

Dr. Thijs E. van Mens, University of Amsterdam

Comments to the Author:

This is a well performed study showing a robust association between multimorbidity and VTE, using relatively unique population data. The methodology is overall valid. However, this reviewers main

concern is the added value of the results to the field. The low granularity and accuracy of these data hamper the interpretability. The findings of multimorbidity, cumulative during a long period of time, being associated with VTE (at any time point during the same period, without a temporal relation), neither adds new insight in epidemiology of VTE, as it is well known that a variety of diseases are associated with VTE. Hence multimorbidity will also be associated. Nor does it have clear clinical implications. This knowledge does not translate into clinical decision in VTE care.

I do not share the conclusion that “ The present study suggests that we might count the total burden of disease into a multimorbidity score to get a better risk estimate of VTE. “ . In my opinion, that would be a step backwards, from predicting using well defined risk factors of which we often also (in part) understand the underlying pathophysiology, to predicting based on a reduced summary of the patient simply because these are (the low-granularity) data that are available.

The research question of familial association is not supported by a clear rationale, except for an explanation that familial association is a proxy for genetic susceptibility.

Author response: We agree that, in the present form, our findings may not be possible to apply in clinical practice and these statements are now deleted from the paper. However, to the best of our knowledge, no study has examined multimorbidity and the severity of multimorbidity (i.e. number of NCDs) and the association with VTE. We now stress that this study is a hypothesis - generating cross-sectional study and that future cohort studies are necessary for establishing a temporal association. Cross - sectional study designs are common in order to generate useful hypotheses that may direct the planning of future case control and cohort studies. The reviewer is correct that we should delete the sentence “The present study suggests that we might count the total burden of disease into a multimorbidity score to get a better risk estimate of VTE” and we have done so. After these edits, we believe that the findings are very interesting and they have not been previously published. We have also added more references such as several systematic reviews and large observational studies in order to put our research into a better context. Previous studies have shown that biological and genetic factors are involved in multimorbidity. The increasing strength of the association with the number of diseases suggests that the present observation is of importance and may reflect something new that is not usually considered in traditional VTE risk factors. For instance, multimorbidity has been associated with increased inflammatory biomarkers and inflammation is a risk factor for VTE. Thus, there is a plausible biological explanation for the association between multimorbidity and severity of multimorbidity and VTE. References for this has also been added and we now discuss this in the discussion section. Moreover, a large UK biobank study has shown that genetic factors are involved in multimorbidity. We have now also added this reference indicating that genetic factors are involved in multimorbidity: Dong G, Feng J, Sun F, Chen J, Zhao XM. A global overview of genetically interpretable multimorbidities among common diseases in the UK Biobank. *Genome Med.* 2021;13(1):110. Thus, there is a clear rationale to study the familial association with multimorbidity because not only VTE but also multimorbidity has a genetic background. The diseases are clearly defined by ICD-10 in the Swedish national patient register, which is a validated register with a high accuracy.

Detailed comments:

In the multivariate models, year of birth was included. Age at diagnoses of VTE seems a more accurate variable.

Author response: Age at diagnosis could not be included as a variable because those who are not affected by VTE will have missing values.

It appears that multimorbidity is determined over the entire study period. This further limits the interpretability, as one would be interested in multimorbidity at time of VTE diagnosis.

The lack of temporal association with VTE is another major drawback, as is the lack absence of data on known life style risk factors. Both limitations are clearly considered and described by the authors.

Author response: We now clearly state and discuss that this is a hypothesis - generating study and that future cohort studies are necessary. However, to the best of our knowledge no researchers have previously studied multimorbidity and the severity of multimorbidity (i.e. number of NCDs) and the association with VTE. We appreciate that this reviewer has acknowledged that we have described the mentioned limitations and hope that our hypothesis-generating study will be of interest for the readers of BMJ Open as well as for the researchers in the field.

Reviewer: 1

Competing interests of Reviewer: None

Reviewer: 2

Competing interests of Reviewer: None

VERSION 2 – REVIEW

REVIEWER	Grainge, Matthew University of Nottingham, Division of Epidemiology and Public Health
REVIEW RETURNED	04-May-2023

GENERAL COMMENTS	I am happy with the change in title to a hypothesis generating study as the acknowledgement that temporality between exposure and outcome has not been established limits the potential for any firm conclusions to be made. I am not sure my suggested edit surrounding Table 1 has been incorporated. It is not clear how this table has changed from the original version.
---

REVIEWER	van Mens, Thijs E. University of Amsterdam
REVIEW RETURNED	26-Apr-2023

GENERAL COMMENTS	The authors have in my view improved the manuscript and toned down the implications of the findings. I have some minor remaining points. The rationale for investigating familial aspect is still not explained explicitly enough in the manuscript. The essence appears to be the explanation in the rebuttal text: "...not only VTE but also multimorbidity has a genetic background." Please explain this more clearly in the background section. Moreover, given the assertion of a genetic component of multimorbidity as well as a link between multimorbidity and VTE, it would perhaps seem appropriate to me to add the patients own multimorbidity score to Model 2 in the familial analysis. This would correct for the multimorbidity-VTE confounding in the familial multimorbidity-VTE correlation. If this erases the correlation, that would suggest that the association of familial multimorbidity runs through the patients own multimorbidity to VTE. Please provide clearer formulation for the "twin score" and the table 5 legend: "Reference are relatives with no diseases (score=0) for score but no or one disease for multimorbidity (≥ 2 score)."
--

	Please explain whether the twins in the study were mono- or dizygotic or state that this information is not available. Minor textual comment: the manuscript now states in 4 places that “ further dissection of the VTE multimorbidity relation is worthwhile”. I suggest more varied phrasing.
--	--

VERSION 2 – AUTHOR RESPONSE

Reviewer: 2

Dr. Thijs E. van Mens, University of Amsterdam

Comments to the Author:

The authors have in my view improved the manuscript and toned down the implications of the findings. I have some minor remaining points.

The rationale for investigating familial aspect is still not explained explicitly enough in the manuscript. The essence appears to be the explanation in the rebuttal text: “...not only VTE but also multimorbidity has a genetic background. “ Please explain this more clearly in the background section.

Author response: In the introduction we now have elaborated a little bit more on the genetics of multimorbidity and the study from UK biobank: “Genetic inheritance has been implicated in multimorbidity.¹⁶ Dong et al have studied 439 common diseases among 385355 patients in UK biobank. They identified 11285 multimorbidities and using genome wide association study (GWAS) data, they showed that 46% of multimorbidities were genetically interpretable (<https://multimorbidity.comp-sysbio.org>).¹⁶”

Moreover, given the assertion of a genetic component of multimorbidity as well as a link between multimorbidity and VTE, it would perhaps seem appropriate to me to add the patients own multimorbidity score to Model 2 in the familial analysis. This would correct for the multimorbidity-VTE confounding in the familial multimorbidity-VTE correlation. If this erases the correlation, that would suggest that the association of familial multimorbidity runs through the patients own multimorbidity to VTE.

Author response: Thanks for this suggestion. It is an interesting idea. However, this may lead to overadjustment. If multimorbidity is inherited multimorbidity will be increased in relatives with familial clustering of multimorbidity. Therefore, this is not done in family studies, i.e. adjusting for disease in the case when disease in proband relative is the predictor. Then most family studies would be negative due to overadjustment as disease in proband and relative are not independent if there is a genetic component.

Please provide clearer formulation for the “twin score” and the table 5 legend: “ Reference are relatives with no diseases (score=0) for score but no or one disease for multimorbidity (≥ 2 score). “ Please explain whether the twins in the study were mono- or dizygotic or state that this information is not available.

Author response. Sorry for being unclear. We have clarified Table 5. Reference are individuals with relatives with no diseases (score=0) for the graded score (score between 1 and ≥ 5). For multimorbidity (score ≥ 2) reference are individuals with relatives with no multimorbidity (score < 2), i.e. no or one disease. Zygosity of twins are not known. In the last sentences in the method in the study design we already have written the following: “Zygosity was assessed only from sex, i.e. twins with

different sex were considered to be DZ twins. We had no access to zygosity data. However, we add a sentence in the table that zygosity of twins not known.

Minor textual comment: the manuscript now states in 4 places that “ further dissection of the VTE multimorbidity relation is worthwhile”. I suggest more varied phrasing.

Author response: Thanks for this comment. We now have varied the phrasing.

Reviewer: 1

Dr. Matthew Grainge, University of Nottingham

Comments to the Author:

I am happy with the change in title to a hypothesis generating study as the acknowledgement that temporality between exposure and outcome has not been established limits the potential for any firm conclusions to be made.

Author response: Thanks for the appreciation of the study.

I am not sure my suggested edit surrounding Table 1 has been incorporated. It is not clear how this table has changed from the original version.

Author response: Sorry for the misunderstanding. We have now changed Table 1 according to the suggestion of reviewer.

Reviewer: 2

Competing interests of Reviewer: none

Reviewer: 1

Competing interests of Reviewer: None declared